# BenchX: A Unified Benchmark Framework for Medical Vision-Language Pretraining on Chest X-Rays

**Yang Zhou[1], Tan Li Hui Faith[2]\*, Yanyu Xu[3], Sicong Leng[4], Xinxing Xu[5],**
**Yong Liu[1]†, Rick Siow Mong Goh[1]†**
[1]Institute of High Performance Computing (IHPC),
Agency for Science, Technology and Research (A*STAR), Singapore
[2]National University of Singapore, Singapore
[3]C-FAIR, Shandong University, China
[4]Nanyang Technological University, Singapore
[5]Microsoft Research Asia Singapore

## Abstract

Medical Vision-Language Pretraining (MedVLP) shows promise in learning generalizable and transferable visual representations from paired and unpaired medical images and reports. MedVLP can provide useful features to downstream tasks and facilitate adapting task-specific models to new setups using fewer examples. However, existing MedVLP methods often differ in terms of datasets, preprocessing, and finetuning implementations. This pose great challenges in evaluating how well a MedVLP method generalizes to various clinically-relevant tasks due to the lack of unified, standardized, and comprehensive benchmark. To fill this gap, we propose BenchX, a unified benchmark framework that enables head-to-head comparison and systematical analysis between MedVLP methods using public chest X-ray datasets. Specifically, BenchX is composed of three components: 1) Comprehensive datasets covering nine datasets and four medical tasks; 2) Benchmark suites to standardize data preprocessing, train-test splits, and parameter selection; 3) Unified finetuning protocols that accommodate heterogeneous MedVLP methods for consistent task adaptation in classification, segmentation, and report generation, respectively. Utilizing BenchX, we establish baselines for nine state-of-the-art MedVLP methods and found that the performance of some early MedVLP methods can be enhanced to surpass more recent ones, prompting a revisiting of the developments and conclusions from prior works in MedVLP. Our code are available at `https://github.com/yangzhou12/BenchX`.

## 1 Introduction

Vision-language pretraining involves training models on large datasets of images and text to learn the relationships between visual and textual data. This pretraining process allows models to learn generalizable representations that can be adapted for specific tasks using fewer training data. Recent advancements in Medical Vision Language Pretraining (MedVLP), driven by rich knowledge from medical reports, play a crucial role in advancing representation learning within the medical domain. By leveraging paired and unpaired medical images and reports, MedVLP has demonstrated strong transfer performance for a wide range of downstream medical tasks with better data efficiency [43, 20, 46].

---

\*Work done during internship at IHPC, A*STAR.
†Joint senior authors.

The success of MedVLP has inspired many pretraining methods in recent years [43, 20, 45, 38, 46, 39, 5, 6]. Despite fruitful MedVLP methods have been proposed, they often use their own evaluation protocols based on varied datasets, customized experimental setups, and diverse training strategies, due to the absence of standard benchmark. To be specific, there exist three main discrepancies preventing fair comparison and systematic analysis of MedVLP methods: 1) Existing MedVLP methods generally utilize diverse datasets or train-test splits for pretraining and finetuning, leading to incomparable results. 2) Prior MedVLP methods adopt custom training strategies and inconsistent data preprocessing such as image resizing and data augmentation, increasing the difficulty of reproducing results and the risk of unfair comparisons. 3) The finetuning protocols of MedVLP methods are often incompatible with each other, due to the heterogeneous model architectures. For example, U-Net [31] is commonly adapted by ResNet [15]-based MedVLP methods for segmentation but is not directly applicable to ViT-based methods. Without a unified finetuning protocol for task adaptation, it is difficult to understand the strengths and weaknesses of different MedVLP methods.

In this work, we aim to address the need of a comprehensive and standard evaluation benchmark for head-to-head comparison and systematical analysis between MedVLP methods using chest X-rays datasets. To facilitate fair and rigorous evaluation, we propose BenchX with the following characteristics:

- **Comprehensive datasets and tasks.** To benchmark the training data, we pretrain MedVLP models on the same training set from the popular MIMIC-CXR dataset [21], and test on nine medical datasets across four tasks.

- **Consistent preprocessing and training.** We develop benchmark suites to standardize data preprocessing and training strategies, which mitigate the impact of inconsistent experimental setups to the MedVLP performance.

- **Unified task adaptation.** We build unified finetuning protocols that accommodate heterogeneous MedVLP methods for consistent task adaptation in classification, segmentation, and report generation, respectively.

Utilizing our BenchX framework, we establish baselines for nine state-of-the-art MedVLP methods. Notably, we observe that with proper training strategies, the performance of some MedVLP models can be improved significantly. In particular, minor adjustments to the classification head and learning rate lead to substantial improvements. For example, ConVIRT [43], one of the first MedVLP methods, shows strong performance when finetuned in the appropriate configuration and becomes competitive with or outperforms more recent approaches such as MedCLIP [38] and MedKLIP [39]. This highlights the unreliability of depending solely on reported results or training without identifying optimal configurations. In general, we observe that MGCA [35] and MRM [46] consistently prove effective. However, the relative performance differences among other MedVLP methods tend to be inconsistent across various tasks. In light of these observations, we advocate for increased attention to the evaluation process in MedVLP. This calls for a revisit of the developments and conclusions from previous works in MedVLP. For reproducibility and extensibility, we will release the code of the whole BenchX framework, all the pre-trained models, config files to reproduce the results, the source information of datasets, and the scripts of preprocessing. We hope that the proposed unified framework contributes to a more robust and reliable evaluation of MedVLP.

## 2 Related Work

**Self-Supervised Pretraining** Self-supervised learning has gained traction as a pretraining paradigm BERT [12], SimCLR [3], MoCO [16]. Unlike traditional supervised methods, self-supervised learning does not rely on ground-truth labels during pretraining. Instead, it leverages self-generated supervision from the data as the training objective. Popular objectives include contrastive learning and masked prediction, both proven effective in capturing complex patterns. Contrastive learning maximizes similarity between embeddings of paired data while minimizing similarity between embeddings of unpaired data. Contrastive Language-Image Pretraining (CLIP) [30], a state-of-the-art method, aligns image-text pairs using a shared embedding space, proving useful in the medical domain. Masked prediction, seen in methods like BERT [12] and Masked Autoencoder (MAE) [17], involves predicting or reconstructing masked parts of original inputs. This approach enhances the model's ability to capture intricate features in image or text data.

**Medical Vision-Language Pretraining**    Traditional vision-language pretraining methods like CLIP [30] excel in general domains but lack specialization in medical images and knowledge. To overcome this limitation, several MedVLP approaches have been proposed recently. Due to the complexity of medical reports, many MedVLP methods aim to improve image-text contrastive (ITC) learning for better alignment [2, 6, 1]. For example, ConVIRT [43] introduces global image-text contrastive learning to align medical images with corresponding reports. GLoRIA [20] and LOVT [27] introduces a local contrastive loss, complementing the global one, to align image patches with words in paired reports. To reduce false negatives in contrastive learning, MGCA [35] additionally performed disease (prototype) level alignment by grouping images and text through clustering. M-FLAG [23] learns to align image embeddings with text by leveraging a frozen language model for training stability and efficiency.

Another branch of approaches uses semantic image-text matching (ITM) losses to encourage the matching between image and text embeddings according to certain semantic labels. MedCLIP [38] uses a semantic matching loss with soft sentence labels to reduce false negatives in contrastive learning and enables both paired and unpaired MedVLP. MedKLIP [39] performs relational triple extraction of medical findings from medical reports and transforms image-text alignment into a classification problem by treating the extracted triples as class labels. KAD [42] improves contrastive learning by sampling positives and negatives according to established medical knowledge bases.

Masked prediction is also frequently used in MedVLP. REFERS [45] combines causal language modeling with image-to-text contrastive learning. Inspired by BERT [12] and MAE [17], MRM [46] uses masked image modelling (MIM) and masked language modelling (MLM) to obtain more informative image representation. PTUnifier [5] proposed a unified architecture that uses prompts to handle various multimodal inputs, taking MLM, ITC, and ITM for pretraining. For a comprehensive review of MedVLP, interesting readers can refer to these very recent survey articles [33, 44].

While all these works have reported promising results, they often conduct incomprehensive comparisons with a few early methods such as ConVIRT [43] and GLoRIA [20]. Additionally, experiments are performed on diverse datasets and tasks, employing different preprocessing and experimental setups for both pretraining and finetuning. Furthermore, in many cases, comparisons are made solely based on the reported results or training without identifying the optimal configurations for the compared methods. These factors make it challenging to enable consistent assessments and systematic analysis of each MedVLP method's strengths and weaknesses.

**Benchmarking Medical Image Pretraining**    There are a few works on benchmarking medical image pretraining. TorchXRayVision [7] is an open-source software library designed for the evaluation of CXR datasets. It provides a common interface for a wide range of publicly available CXR datasets, offering pretrained classification and representation learning models as baselines or feature extractors. ViLMedic [9], on the other hand, is a modular framework for multimodal medical tasks. It implements several baseline methods for medical visual question answering, radiology report generation, and pretraining.

Despite the existence of these remarkable frameworks, there remains a significant gap in benchmarking MedVLP methods. TorchXRayVision focuses on vision tasks and does not consider multimodal data and MedVLP. While ViLMedica builds the codebase to implement a few MedVLP baselines (such as ConVIRT [43] and GLoRIA [20]) and perform multimodal tasks (such as visual question answering and report generation), it does not address the discrepancies in adapting different MedVLP methods to unified task adaption pipelines. Additionally, it does not conduct evaluations across a wide range of existing MedVLP methods and downstream tasks for a consistent and comprehensive comparison. In this work, we aim to close this gap and contribute to the larger landscape of MedVLP benchmarking.

**Comparison with Existing Works**    BenchX stands out from existing benchmark frameworks such as TorchXRayVision and ViLMedic by addressing key gaps in the evaluation and comparison of MedVLP methods:

- **Comprehensive multimodal benchmarking.** Unlike TorchXRayVision, which focuses only on vision tasks, BenchX offers a unified framework for benchmarking MedVLP methods across both vision and language tasks, enabling the evaluation of multimodal models.

- **Standardized task adaptation protocol.** BenchX introduces a consistent and standardized task adaptation pipeline, addressing a gap present in ViLMedic. This standardization reduces variability due to differing implementation details, ensuring fair comparisons across different MedVLP methods.
- **Diverse dataset and benchmark suite.** BenchX includes a comprehensive dataset and a diverse set of benchmarks, overcoming limitations in existing frameworks. This diversity enables more robust and reliable evaluations of MedVLP methods, establishing new baselines and promoting advancements in the field.

## 3  The BenchX Framework

In this section, we introduce BenchX, a unified benchmark framework designed for head-to-head comparison and systematic evaluation of MedVLP methods. Our objective is to standardize preprocessing, pretraining, and finetuning and establish unified evaluation protocols to accommodate heterogeneous MedVLP methods with minimal customization. This ensures that the performance of downstream tasks is primarily determined by MedVLP methods, without being influenced by custom experimental setups.

### 3.1  Benchmarking Training Datasets

To conduct a comprehensive robust evaluation of MedVLP methods, we leverage multiple publicly available datasets. Detailed information regarding the datasets, including specific names, sources, and characteristics, will be provided in the supplementary material.

**Datasets for Pretraining:**  The MIMIC-CXR dataset [21] comprises over 370,000 check X-rays (CXRs) from more than 220,000 patient studies, serving as a prominent pretraining dataset for numerous MedVLP methods. However, inconsistencies arise in the preprocessing of MIMIC-CXR across different methodologies. For instance, MGCA [35] and PTUnifier [5] only utilize frontal chest radiographs, while MedCLIP [38] and MRM [46] use both frontal and lateral views for pretraining. Various MedVLP methods, including GLoRIA [20], MedCLIP [38], and PTUnifier [5], opt for different or additional pretraining datasets. To robustly assess the effectiveness of MedVLP methods within a standardized setting, we exclusively pretrain MedVLP models on frontal images derived from the official training split of the MIMIC-CXR dataset [21].

**Datasets for Downstream Tasks:**  We benchmark the performance of MedVLP methods across four downstream tasks including classification, segmentation, report generation, and image-text retrieval using nine public chest X-ray datasets. For classification, we use two multilabel classification datasets: NIH ChestX-ray [37] and VinDr [28], and three binary classification datasets: COVIDx CXR4 [36], RSNA [32], and SIIM [41]. For segmentation, we test on Object CXR [18], RSNA [32], SIIM [41], and TBX11k [24]. For report generation, we employ the IUXray [10] dataset. For image-text retrieval, following GLoRIA [20] and MedCLIP [38], we retain partial data to construct the MIMIC5x200 dataset, where we sample 200 image-text pairs for each of the medical finding from Atelectasis, Cardiomegaly, Edema, Pleural, and Effsion.

> **Remark #1:** To fully understand the effectiveness of MedVLP strategies, it is crucial to apply the same pretraining and finetuning dataset configuration to all the compared methods, which is usually ignored by prior works. By collecting nine datasets across four medical tasks, our BenchX framework enables extensive evaluation in terms of the transferability of learned representations.

### 3.2  Standardizing Data Preprocessing and Training Strategies

**Data Preprocessing:**  Follow common data transforms, we first resize input images to 256x256 and then apply random crop to 224x224 for data augmentation for classification and report generation. We simply resize input images to 512x512 without further preprocessing.

**Training Strategies:**  Although classification is a standard task tested by all the MedVLP methods, we find that their implementations have subtle yet crucial differences, which could significantly

affect the performance. To achieve the best performance for each MedVLP method, we explore three key strategies beyond naive training: 1) Applying layer normalization before feeding image embeddings into the classifier; 2) Initializing the classifier with values drawn from a truncated normal distribution, and 3) Applying discriminative learning rates for the image encoder and classifier. As will be demonstrated in the experiments, these training strategies can substantially enhance the performance of certain MedVLP methods.

> **Remark #2:** Despite not being explicitly emphasized in the literature, some MedVLP methods have utilized the above strategies to obtain advantage over others. To avoid unfair comparison and inconclusive results, we empirically determine the optimal training strategies when obtaining the experimental results for each method.

## 3.3 Unifying Task Adaptation Pipelines

As MedVLP methods generally are not pretrained for certain downstream tasks, they need to introduce a task-specific head during the finetuning stage. Such task-specific head can range from a simple linear layer for classification to a more complicated network like U-Net [31] for segmentation. Due to heterogeneous MedVLP model architectures, the heads used by different MedVLP methods can be incompatible with each other. This results in inconsistent evaluation when it comes to comparing the performance of CNN-based Med-

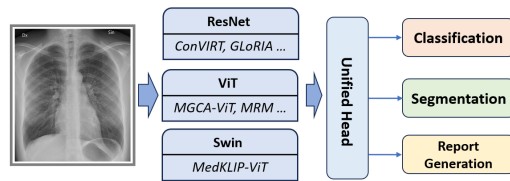

Figure 1: The illustrative tasks adaptation pipeline.

VLP methods with ViT-based ones. To ensure consistent task adaptation, we propose the following unified pipelines for classification, segmentation, and report generation, respectively.

**Classification:** For assessing the classification performance of MedVLP methods, we follow most existing works to add a linear classifier on top of the image encoder and adapt the MedVLP model in a full finetuning setting, where both the image encoder and the linear classifier will be updated. Following [20, 35, 46], we employ varying amounts of training data (1%, 10%, or 100%) to evaluate data efficiency.

**Segmentation:** Adapting MedVLP models for segmentation involves integrating the image encoder into certain segmentation networks. However, determining the segmentation network for benchmarking is nontrivial, as adapting MedVLP models to specific segmentation networks may not always be feasible. For instance, the widely used U-Net is generally incompatible with ViT-based MedVLP models. Without a unified finetuning protocol, it is unclear that whether improved performance is attributed to superior MedVLP methods or a more capable segmentation network.

To address this challenge, we propose a unified segmentation pipeline by adapting the implementation of UperNet [40] from the open-source mmsegmentation package [8]. UperNet is a versatile segmentation model compatible with various backbone architectures, including ResNet [15], ViT [13], Swin Transformer [26], and more. This model allows minimal modifications when switching from one MedVLP model to another. Following the approach in GLoRIA [20] and MGCA [35], we finetune UperNet with a frozen backbone from the pretrained MedVLP image encoder. This simplifies the training process and makes the segmentation performance depend more on the ability of MedVLP methods in representation learning.

**Report Generation:** Similar to segmentation, we adapt R2Gen [4] as the head for report generation, with the image encoder frozen from a specified MedVLP model. R2Gen is chosen for its simplicity and adaptability in supporting various image encoders. It is noteworthy that there is a prevalent trend of combining MedVLP models with large language models (LLM) [34], leading to state-of-the-art performance in report generation. However, since it is out of the scope of this paper, the exploration of LLM-based report generation is deferred to future work.

Table 1: Multi-label classification performance (%) of MedVLP methods (**Best**, Second Best).

| Model | NIH (AUROC) | | | VinDr (AUROC) | | |
| --- | --- | --- | --- | --- | --- | --- |
| | 1% | 10% | 100% | 1% | 10% | 100% |
| ConVIRT | 77.0±0.1 | 81.5±0.01 | 84.2±0.06 | 88.1±0.1 | 90.5±0.1 | 90.9±0.2 |
| GLoRIA | 74.2±0.5 | 81.0±0.16 | 83.8±0.15 | 87.5±0.1 | 90.3±0.2 | 91.3±0.1 |
| MedCLIP-R50 | 74.2±0.6 | 79.5±0.36 | 83.9±0.08 | 83.0±2.0 | 87.7±0.3 | 89.8±0.4 |
| MedCLIP-ViT | 76.1±0.3 | 81.4±0.25 | 84.5±0.17 | 83.6±1.5 | 89.7±0.5 | 88.7±0.4 |
| MedKLIP | 75.2±0.1 | 80.3±0.08 | 83.9±0.08 | 77.5±1.9 | 85.8±2.1 | 89.9±0.5 |
| M-FLAG | 66.5±0.5 | 78.4±0.55 | 84.0±0.04 | 69.2±2.1 | 81.7±0.8 | 86.6±0.9 |
| MGCA-R50 | 73.2±0.3 | 79.9±0.08 | 83.5±0.04 | 84.5±0.5 | 89.1±0.3 | 90.6±0.2 |
| MGCA-ViT | 78.2±0.1 | 82.4±0.03 | 84.4±0.05 | **88.3±0.1** | **91.5±0.2** | **91.8±0.3** |
| MRM | **80.1±0.1** | **83.5±0.10** | **85.3±0.05** | 87.1±0.1 | 89.9±0.1 | 91.2±0.3 |
| REFERS | 76.4±0.3 | 81.3±0.01 | 83.7±0.06 | 87.1±0.1 | 89.4±0.3 | 90.0±0.5 |

> **Remark #3:** Our BenchX is designed to unify the evaluation protocol for each downstream task, ensuring that the performance of compared methods primarily depends on their ability in representation learning rather than individual task-specific adaptations. By doing so, we eliminate unnecessary task or model-specific modifications, preventing our framework from favoring certain MedVLP methods over others.

## 4 Benchmark Results

In this section, we utilize the propose framework to benchmark the performance of MedVLP methods. For each experiment, we report the average results and the standard deviation from three independent runs with different random seeds. The implementation details can be found in the supplementary material.

**Compared Methods:** We evaluate nine state-of-the-art MedVLP methods, including ConVIRT [43], GLoRIA [20], MedCLIP [38], MedKLIP [39], M-FLAG [23], MGCA [35], MRM [46], PTUnifier [5], and REFERS [45]. These models are originally pretrained on diverse datasets, based on heterogeneous architectures such as ResNet [15], Vision Transformer (ViT) [13], Swin Transformer [26], and custom models, and combined with varied text encoders including BERT [12] and its biomedical variants such as ClinicalBERT [19], CXR-BERT [2], and BioMed ROBERTa [14]. We follow the official implementation of each MedVLP method to pretrain MedVLP models on the same training set defined in Section 3.1. We also test the released checkpoints (if available) of the compared MedVLP methods and verify that our pretrained models achieve similar performance with the released ones in our experiments. For MedCLIP [38], we can only evaluate its released checkpoints, since training MedCLIP requires dedicated sentence labels that are not publicly available.

### 4.1 Medical Image Classification

We first assess the classification performance of MedVLP methods. The evaluation metrics include the area under the ROC curve (AUROC) for multilabel classification, measuring the model's ability to differentiate between true positives and false positives across various threshold values. For binary classification, we employ F1, the harmonic mean of precision and recall, because we find that AUROC may not fully reflect the performance difference across MedVLP methods.

Tables 1 and 2 presents the classification results using different percentages of training samples. When comparing across multiple datasets, it appears that no single method consistently outperforms others. However, MedCLIP-ViT, MGCA-ViT, and MRM stand out as the top-performing methods, achieving the top two performances in most cases. Across all datasets except for SIIM, MedVLP methods trained with 10% of data yield results similar to those trained with 100% of data, demonstrating the effectiveness of MedVLP in providing good data efficiency for downstream tasks. MedCLIP and MGCA have both ResNet and ViT-based implementations. From the experimental results, ViT-based methods generally outperform their ResNet-based counterpart. This finding is consistent with prior results in MedCLIP [38] and MGCA [35].

Table 2: Binary classification performance (%) of MedVLP methods (**Best**, Second Best).

| Model | COVIDx (F1) | | | SIIM (F1) | | | RSNA (F1) | | |
|---|---|---|---|---|---|---|---|---|---|
| | 1% | 10% | 100% | 1% | 10% | 100% | 1% | 10% | 100% |
| ConVIRT | 67.4±0.6 | 68.7±0.1 | 68.1±0.1 | 62.8±0.7 | 64.8±1.7 | 72.8±0.8 | 58.0±0.5 | 63.3±0.3 | 65.0±0.8 |
| GLoRIA | 66.6±0.6 | 68.2±0.1 | 68.3±0.0 | 59.3±1.0 | 63.4±1.1 | 69.0±2.3 | 60.1±0.6 | 62.0±1.1 | 64.7±1.0 |
| MedCLIP-R50 | **68.5±1.7** | 68.3±0.2 | 68.3±0.1 | 64.8±1.1 | 68.4±1.1 | 73.2±1.7 | 62.9±0.5 | 63.9±0.3 | 65.3±0.8 |
| MedCLIP-ViT | 67.1±0.5 | 68.7±0.4 | 68.3±0.1 | **68.6±0.8** | 71.5±1.1 | **75.7±0.2** | 63.5±0.5 | 65.3±1.0 | 66.2±0.8 |
| MedKLIP | 66.5±0.2 | **69.3±0.6** | 68.3±0.3 | 61.4±0.3 | 64.4±2.1 | 72.7±1.4 | 60.4±0.6 | 61.9±1.4 | 66.0±0.6 |
| M-FLAG | 67.6±0.3 | 69.2±1.0 | 68.1±0.1 | 47.1±0.3 | 61.8±1.5 | 72.1±1.6 | 56.0±0.9 | 60.3±1.4 | 64.4±0.3 |
| MGCA-R50 | 68.2±1.1 | 68.4±0.2 | 68.0±0.1 | 59.7±1.2 | 61.3±1.0 | 69.4±0.8 | 57.3±0.5 | 61.9±0.6 | 64.0±1.3 |
| MGCA-ViT | 66.5±0.9 | 68.1±0.1 | 68.2±0.0 | 66.3±0.3 | 68.6±0.9 | 73.3±0.8 | 61.0±1.3 | 64.3±0.4 | 66.9±1.4 |
| MRM | 67.4±0.6 | 68.2±0.4 | 68.3±0.2 | 65.0±0.5 | 69.3±1.0 | 75.6±0.7 | 62.6±1.1 | **66.6±0.3** | 66.5±0.2 |
| REFERS | 66.7±0.0 | 66.6±1.0 | **68.5±0.8** | 60.8±1.0 | 66.9±0.7 | 72.6±0.3 | 61.7±0.7 | 63.8±0.1 | **67.2±0.3** |

Table 3: Segmentation performance (%) in mDice score (**Best**, Second Best).

| Method | Obj-CXR | RSNA | SIIM | TBX11K |
|---|---|---|---|---|
| ConVIRT | 79.82±0.59 | 74.72±0.12 | 76.02±0.44 | 84.98±0.59 |
| GLoRIA | 77.23±0.13 | 74.41±0.41 | 73.39±0.43 | 83.17±0.36 |
| MedCLIP-R50 | 79.88±0.23 | 75.45±0.11 | 76.35±0.44 | 85.52±0.17 |
| MedCLIP-ViT | 79.64±0.35 | 73.29±1.41 | 76.48±0.38 | 85.62±0.07 |
| MedKLIP | 78.17±0.29 | 74.68±0.42 | 77.78±0.69 | 87.06±0.31 |
| M-FLAG | 73.96±0.30 | 67.86±0.63 | 68.13±0.75 | 79.12±0.16 |
| MGCA-R50 | 80.27±0.07 | 75.04±0.59 | 77.04±0.48 | 87.05±0.19 |
| MGCA-ViT | **81.68±0.26** | 75.48±0.28 | 77.22±0.51 | 86.89±0.39 |
| MRM | 80.45±0.02 | **75.69±0.56** | **78.66±0.52** | **87.85±0.47** |
| PTUnifier | 80.64±0.10 | 74.54±0.50 | 74.91±0.58 | 85.78±0.05 |
| REFERS | 80.47±0.08 | 75.52±0.34 | 75.33±0.85 | 86.39±0.26 |

Surprisingly, ConVIRT, one of the first MedVLP methods, demonstrates good performance, which is on par with or even superior to many state-of-the-art methods, including GLoRIA, MedCLIP, and REFERS in certain cases. This can be attributed to the refined training strategies introduce in Section 3.2. Notably, these training strategies enhance the performance not only for ConVIRT but also for other MedVLP methods. The impact of training strategies will be discussed in detail in Section 4.6.

## 4.2 Medical Image Segmentation

We then evaluate the effectiveness of MedVLP methods for medical image segmentation. Table 3 shows the segmentation results on the Obejct CXR, RSNA, SIIM, and TBX11K [25] datasets, where we use the mean Dice scores (mDice) as the evaluation metric and highlight the best and second best results. Similar to the classification results, no method consistently achieves the best performance. However, MRM stands out as the overall best method, achieving the best results on the RSNA, SIIM, and TBX11K datasets. This implies that the combination of masked image and language modeling proposed in MRM [46] may be beneficial for the segmentation tasks.

MGCA [35] achieves top 2 performance on the Object CXR and RSNA datasets, which demonstrate the effectiveness of the crossmodal prototype alignment strategy proposed in MGCA. MedKLIP [39] generally performs well on the SIIM and TBX11K datasets, which indicates that MedVLP can benefit from careful information extraction from medical reports. Notably, ConVIRT performs reasonably well on all datasets and obtains better results than more recent methods such as GLoRIA, PTUnifier, and M-FLAG in many cases. In particular, we notice that the results of ConVIRT obtained from our benchmark framework greatly outperform the reported ones in [20, 35]. This suggests that some early work in the field of MedVLP may need to be revisited.

Here is the table with the results converted to percentages, including the standard deviations, and the ± replaced with ±:

Table 4: Radiology report generation results on the IUXray dataset (**Best**, Second Best).

| METHOD | BLEU1 | BLEU2 | BLEU3 | BLEU4 | ROUGEL | METEOR |
|---|---|---|---|---|---|---|
| BASELINE | 41.5±4.7 | 25.6±3.0 | 17.9±2.3 | 13.3±1.8 | 32.9±1.9 | 16.5±2.2 |
| CONVIRT | 44.3±1.7 | 28.6±1.3 | 20.1±0.8 | 14.8±0.6 | 36.8±1.3 | 18.7±0.7 |
| GLORIA | 46.6±5.2 | **31.6±2.8** | **22.7±1.7** | **17.0±1.1** | **38.7±0.7** | **20.2±1.0** |
| MEDCLIP-R50 | 44.0±3.1 | 29.5±1.3 | 21.6±0.7 | 16.3±0.6 | 38.0±1.0 | 18.9±0.6 |
| MEDCLIP-VIT | 42.1±4.6 | 28.0±3.2 | 20.1±2.6 | 15.1±2.0 | 38.2±1.1 | 18.0±0.9 |
| MEDKLIP | **47.0±1.1** | 31.0±2.2 | 22.2±2.1 | 16.7±1.6 | 37.9±0.9 | 19.4±0.5 |
| PTUNIFIER | 46.8±2.2 | 30.7±1.9 | 21.7±1.1 | 16.2±0.7 | 38.0±0.6 | 19.4±1.1 |
| M-FLAG | 41.2±2.9 | 27.4±2.4 | 19.6±1.9 | 14.7±1.6 | 37.1±0.9 | 18.5±0.4 |
| MGCA-R50 | 45.7±3.3 | 30.0±2.7 | 21.3±1.8 | 15.9±1.4 | 37.5±1.6 | 19.1±1.3 |
| MGCA-VIT | 46.2±3.4 | 31.1±3.1 | 22.5±2.6 | 17.0±2.1 | 38.4±1.9 | 19.5±1.0 |
| MRM | 44.5±5.5 | 30.8±3.4 | 22.3±2.4 | 16.5±1.7 | 38.1±1.3 | 19.0±0.8 |
| REFERS | 46.6±2.2 | 30.5±0.9 | 21.6±0.9 | 16.1±0.9 | 37.7±0.7 | 19.5±0.2 |

## 4.3 Radiology Report Generation

Next, we explore the effectiveness of MedVLP in radiology report generation. The task is to generate a medical report that correctly describe the medical findings in a given image. In addition to the finetuned MedVLP models, we also compare with a baseline by training G2Gen from scratch, using ResNet50 pretrained on natural images as the image encoder. We use natural language generation (NLG) metrics such as BLEU [29], METEOR [11], and ROUGE-L [22] to assess the performance of report generation. The BLEU score measures the similarity between the generated and reference reports based on the precision of n-grams (words) in the generated text. ROUGE-L measures the longest common subsequence between the generated output and the reference report. METEOR assesses the overall generation quality by considering precision, recall, and alignment between the generated text and ground truth.

Table 4 shows the results of report generation on the IUXray dataset. As can be seen, all the MedVLP methods exhibit significantly better results than the baseline, demonstrating the effectiveness of MedVLP in improving report generation. On the other hand, the performance difference is marginal across all the MedVLP methods. This is probably because the performance of report generation is mainly determined by the generation head rather than the pretrained image encoder. Among all the MedVLP methdos, GLoRIA obtains the best results in all metrics except for BLEU1, and MGCA-ViT is generally the second best method. This could be attributed to their design to align both global and local embeddings between images and reports.

## 4.4 Medical Image-Text Retrieval

In this section, we conduct experiments on our MIMIC 5x200 dataset to evaluate MedVLP methods for image-text retrieval in the zero-shot setting. Given an image as input query, the task is to find the matched reports by computing the similarity between the query image and all candidate reports using the learned representations. Considering retrieval performance can be influenced by factors beyond the image encoders, benchmarking MedVLP methods in image-text retrieval could be tricky. Nevertheless, we provide the results for

Table 5: Image-text retrieval results on the MIMIC 5x200 datasets (**Best**, Second Best).

| Model | H@1 | H@5 | H@10 | P@1 | P@5 | P@10 |
|---|---|---|---|---|---|---|
| ConVIRT | 61.9 | 88.2 | 94.2 | 61.9 | 54.9 | 52.5 |
| GLoRIA | 54.6 | 86.3 | 93.6 | 54.6 | 49.7 | 47.2 |
| MedCLIP-R50 | 16.1 | 35.1 | 46.4 | 16.1 | 16.6 | 18.8 |
| MedCLIP-ViT | 42.0 | 77.9 | 88.8 | 42.0 | 41.0 | 40.6 |
| MGCA-R50 | 57.9 | 87.9 | 95.8 | 57.9 | 53.0 | 50.2 |
| MGCA-ViT | 63.3 | 90.4 | 95.5 | 63.3 | **56.4** | **52.6** |
| PTUnifier | **78.7** | **99.5** | **100.0** | **78.7** | 38.4 | 23.4 |
| REFERS | 54.4 | 83.4 | 90.5 | 54.4 | 52.5 | 50.5 |

comprehensiveness. It is worth noting that only contrastive learning-based methods are applicable to the image-text retrieval task. Consequently, we exclude other MedVLP methods, such as MRM, MedKLIP, and M-FLAG, from our comparison.

Table 5 shows the retrieval results on the MIMIC 5x200 dataset. We use HiT@K and Precision@K as the performance metrics, which measure the presence and proportion of correct reports among the

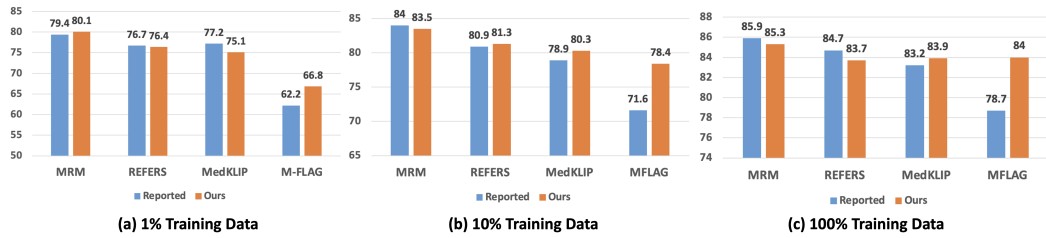

Figure 2: Comparison with reported results (AUROC) on the NIH dataset.

top K predictions, respectively. The results demonstrate that PTUnifer achieves the highest Hit@K value, surpassing the second-best method by a significant margin. This success is likely attributable to image-text matching loss used by PTUnifier during pretraining, which makes PTUnifier easy to query matched reports from a given image. However, PTUnifer exhibits relatively low P@K scores, which means that means that reports describing the same disease do not frequently appear among the top K predictions of PTUnifer. This suggests that the text representations learned by PTUnifer do not align well semantically with the image embeddings. In terms of Precision@K, MGCA-ViT emerges as the top-performing method, while ConVIRT also yields comparable results. This observation suggests that ConVIRT may be more effective than previously believed.

## 4.5 Comparison with Reported Results

To verify the fidelity of our benchmarking study, we compare our experimental results with those reported on the NIH dataset for MedKLIP, M-FLAG, MRM, and REFER. These methods were originally tested under the same experimental setup with our benchmark framework. In Figure 2, the classification results obtained by our BenchX framework are comparable and sometimes superior to the reported results, thereby demonstrating the fidelity of our benchmarking study.

## 4.6 Impact of Training Strategies

Finetuning MedVLP models is generally nontrivial even for simply tasks like linear classification. Naive implementation of introducing linear layer as the classification head is insufficient to unleash the power of MedVLP models in finetuning. In addition, due to the heterogeousity of MedVLP models, one may need to tune the finetune protocol to achieve the best performance for each MedVLP methods.

To exemplify this problem, we explore three common training strategies to improve the naive classification implementation including Layer Normalization (LN), Truncated Normal Initialization

Table 6: Classification results (AUROC) with different training strategies on the NIH dataset with 1% training data.

|  | None | +DLR | +DLR+LN | All |
|---|---|---|---|---|
| ConVIRT | 71.7 | 76.9 (↑) | 74.5 (↓) | 77.0 (↑) |
| GLoRIA | 72.8 | 74.2 (↑) | 70.6 (↓) | 74.9 (↑) |
| MedCLIP-R50 | 74.1 | 73.7 (↓) | 74.2 (↑) | 73.8 (↓) |
| MedCLIP-ViT | 75.5 | 75.7 (↑) | 75.9 (↑) | 70.7 (↓) |
| MedKLIP | 74.4 | 71.9 (↓) | 75.2 (↑) | 73.7 (↓) |
| MGCA-R50 | 72.8 | 73.0 (↑) | 69.6 (↓) | 73.8 (↑) |
| MGCA-ViT | 77.7 | 78.1 (↑) | 78.2 (↑) | 78.2 (=) |
| MRM | 77.9 | 80.0 (↑) | 79.5 (↓) | 80.1 (↑) |
| REFERS | 76.8 | 75.9 (↓) | 76.2 (↓) | 75.6 (↓) |

(TNI), and Discriminative Learning Rates (DLR), which have been introduced in Section 4.6. Table 6 show the impact of each training strategy on the classification performance. We find applying TNI alone only leads to worse results and opt not to report these results for simplicity.

As shown, naively finetuning a learn classifier (Column "None") leads to suboptimal results for all the MedVLP methods except REFERS. On the other hand, applying certain refinements generally improve the classification performance, while the best configuration varies for different MedVLP methods. Notably, the performance of ConVIRT boosts from 71.7% to 77.0% and becomes the third best method when applying LN, TNI, and DLR simultaneously (Column "All"). This demonstrates the complexity in bencharmking MedVLP methdos, and suggests that more comprehensive parameter search should be made for fair and consistent comparison.

Table 7: Overall performance (%) of each MedVLP method across different tasks (**Best**, Second Best).

| Method | M-CLS (AUC)↑ | B-CLS (F1)↑ | SEG (mDice)↑ | RRG (BLEU4)↑ | Avg. Rank↓ |
|---|---|---|---|---|---|
| ConVIRT | 85.37 | 65.56 | 78.89 | 14.8 | 6.38 |
| GLoRIA | 84.68 | 64.06 | 77.05 | **17.0** | 5.88 |
| MedCLIP-R50 | 83.02 | 67.17 | 79.80 | 16.3 | 5.25 |
| MedCLIP-ViT | 84.00 | **68.33** | 78.76 | 15.1 | 5.75 |
| MedKLIP | 82.77 | 65.56 | 79.42 | 16.7 | 6.13 |
| M-FLAG | 77.73 | 62.96 | 72.77 | 14.7 | 10.00 |
| MGCA-R50 | 83.47 | 64.69 | 79.85 | 15.9 | 6.50 |
| MGCA-ViT | 86.10 | 67.03 | 80.32 | **17.0** | 2.38 |
| MRM | **86.18** | 67.72 | **80.66** | 16.5 | **2.00** |
| REFERS | 84.65 | 66.06 | 79.93 | 16.1 | 4.75 |

## 4.7 Overall Performance

Finally, we summarize the overall performance of the compared MedVLP methods across various tasks, including multi-label classification (M-CLS), binary classification (B-CLS), segmentation (SEG), and radiology report generation (RRG). We exclude the medical image-text retrieval task from this comparison because not all MedVLP models support it. Table 7 presents the average results using 100% of the training data across all datasets for each task, along with the average ranking of each method across the four tasks. The experimental results demonstrate that MRM and MGCA-ViT consistently achieve strong performance and outperform other methods across multiple tasks. Other recent MedVLP models such as MedCLIP, MedKLIP, and REFERS generally outperform earlier approaches such as ConVIRT and GLoRIA, but the improvements are not as substantial as reported. This indicates that while significant progress has been made in MedVLP, the reported results in current research may not fully capture the optimal performance of certain baseline MedVLP methods. This finding calls for a reevaluation of the effectiveness of existing methods and the conclusions drawn from them.

## 4.8 Limitations

Our BenchX framework has several limitations: 1) In this work, we focus on benchmarking MedVLP methods in terms of the performance of the *pretrained image encoder* on selective downstream tasks such as classification and segmentation. More studies on crossmodal tasks such as vision question answering are needed to fully understand the effectiveness of MedVLP methods. 2) We conduct experiments on public check X-rays datasets to facilitate comparisons with existing works, while the applications of MedVLP methods are not limited to check X-Rays. 3) The focus of this study is to compare MedVLP methods in a unified experimental setup with minimal individual modifications. Although we have verified that our experimental results are comparable to the reported ones, it is still possible that some methods may achieve suboptimal results due to incomplete search of hyper-parameters or model configurations.

## 5   Conclusion

We have introduced BenchX, a unified benchmark framework designed to facilitate head-to-head comparison and the systematic evaluation between MedVLP methods by mitigating the impact of non-standard experimental setups to the MedVLP performance. Our framework allows various MedVLP methods to be adapted for downstream tasks in a unified pipeline, addressing discrepancies among MedVLP methods in downstream evaluations. Through an extensive study on four typical downstream medical tasks, we established baselines for nine MedVLP methods across nine medical datasets. We observe that finetuning strategies could substantially influence the performance of downstream tasks. Different MedVLP methods often require specific training configurations to achieve the best performance due to the heterogeneity of MedVLP models. In light of these observations, we advocate for increased attention to the evaluation process and prompt a revisit of the developments and conclusions from previous works in MedVLP. One of the key features of BenchX is its extensibility. It has supported many existing models with various architectures. One can easily adapt it to new models and integrate new datasets, allowing for continuous expansion and improvement. We believe this work will be a useful tool and could facilitate research in medical vision-language pre-training.

## Acknowledgements

This research is supported by the National Research Foundation Singapore under the AI Singapore Programme (AISG Award No: AISG2-TC-2023-013). This work is also supported in part by the Agency for Science, Technology and Research (A*STAR) through its AME Programmatic Funding Scheme under Project A20H4b0141.

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

Table 8: Statistics of the test datasets.

| DATASET | IMAGE SIZE | DATASET SIZE | TASK | ANNOTATION |
|---------|-----------|-------------|------|-----------|
| NIH CHESTX-RAY 14 | $224 \times 224$ | 112,120 | CLS | 14 CLASSES |
| VINDR-CXR | $512 \times 640$ | 18,000 | CLS | 28 CLASSES, BBOXES |
| COVIDX CXR-4 | $1024 \times 1024$ | 84,818 | CLS | 2 CLASSES |
| SIIM-ACR PTX | $512 \times 512$ | 12,047 | CLS, SEG | 2 CLASSES, MASKS |
| RSNA PNEUMONIA | $1024 \times 1024$ | 26,684 | CLS, SEG | BBOXES |
| IU-XRAY | $512 \times 640$ | 3,955 | RRG | IMAGE-REPORT PAIRS |
| OBJECT CXR | $2048 \times 2624$ | 10,000 | DET | BBOXES, ELLIPSE, POLYGONS |
| TBX11K | $512 \times 512$ | 11,200 | CLS, SEG | 3 CLASSES, BBOXES |
| MIMIC 5x200 | $512 \times 512$ | 1,000 | RET | IMAGE-REPORT PAIRS |

# A  Appendix

## A.1  Test Datasets

We evaluate MedVLMs on 9 public datasets across 4 tasks including classification (CLS), report generation (RRG), segmentation (SEG), and image-text retrieval (RET). All the experiments are conducted on a Nvidia A100 GPU and the time for each experimental run is up to three hours. Table 8 provided the statistics for the tested datasets. The detailed dataset information is as follows:

- NIH ChestX-ray [37] consists of 112,120 frontal-view CXRs with 14 disease labels from 30,805 unique patients. To make our results comparable with those reported by existing works, we follow [45, 46] to use the same training, validation, and test split corresponds to 70%, 10%, and 20% of the entire dataset, respectively.

- VinDr-CXR [28] contains more 18,000 CXRs collected from two major hospitals in Vietnam, where each image is annotated with both class labels and bounding boxes for 28 findings or diseases. We use the official data split with the training set of 15,000 images and the test set of 3,000 images, respectively. We further randomly selected 3,000 images from the training set to construct a validation set for parameter selection. Therefore, the final training, validation, and test sets contain 12,000, 3,000, and 3,000 samples, respectively.

- COVIDx-CXR4 [36] consists of 84,818 images from 45,342 subjects for COVID-19 detection, which is a binary classification task. We employ the official data split corresponds to 80%, 10%, and 10% of the entire dataset, respectively.

- SIIM-ACR Pneumothorax Segmentation (SIIM) [41] is designed to support the development of segmentation models for identifying pneumothorax in CXRs. SIIM contains 12,047 frontal-view CXRs with mask annotations of pneumothorax. Following [20], we adopt the same training, validation, and test split, where each constitutes 70%, 15%, and 15% of the entire dataset, respectively.

- RSNA Pneumonia [32] contains 26,684 images with mask annotations of pneumonia. We build the data split corresponds to 70%, 15%, and 15% of the entire dataset, respectively.

- IU X-RAY [10] consists of 7,470 chest X-ray images and 3,955 reports. We follow [4] to exclude the samples without reports and use the same training, validation, and test split corresponds to 70%, 10%, and 20% of the entire dataset, respectively.

- Object CXR [18] contains 10,000 frontal-view CXRs with annotations of foreign objects, where 5,000 CXRs have foreign objects and the other 5,000 CXRs have no foreign object. We use the official data split with the training, validation, and test sets consisting of 8,000, 1,000, and 1,000 images, respectively.

- TBX11K [24] consists of 11,200 X-rays with bounding box annotations for tuberculosis (TB) areas, where there are 5,000 healthy cases, 5,000 sick but non-TB cases, and 1,200 cases with manifestations of TB. We use the official data split with the training, validation, and test sets consisting of 6,600, 1,800, and 2,800 samples, respectively.

Table 9: Selected hyper-parameters per method on the NIH dataset.

| Method | Learning Rate | Batch Size | Optimizer | LN | DLR |
|---|---|---|---|---|---|
| ConVIRT | $1 \times 10^{-4}$ | 64 | Adam | Yes | Yes |
| GLoRIA | $1 \times 10^{-4}$ | 64 | Adam | Yes | Yes |
| MedCLIP-R50 | $1 \times 10^{-5}$ | 64 | Adam | No | No |
| MedCLIP-ViT | $1 \times 10^{-5}$ | 32 | Adam | No | No |
| MedKLIP | $1 \times 10^{-4}$ | 128 | Adam | No | Yes |
| M-FLAG | $1 \times 10^{-4}$ | 32 | Adam | Yes | No |
| MGCA-R50 | $1 \times 10^{-5}$ | 32 | Adam | Yes | No |
| MGCA-ViT | $1 \times 10^{-2}$ | 64 | SGD | Yes | Yes |
| MRM | $3 \times 10^{-2}$ | 64 | SGD | Yes | Yes |
| REFERS | $3 \times 10^{-2}$ | 32 | SGD | Yes | No |

- We follow [20, 6] to construct MIMIC 5x200 to detect 5 diseases including Atelectasis, Cardiomegaly, Edema, Pleural, Effsion by randomly sampling 200 exclusive samples for each class from the MIMIC-CXR dataset.

## A.2 Implementation Details

- Overall Setup: Each experiment is run three times with different random seeds, and the average results are reported. We monitor performance on the validation set at each epoch and select the best checkpoint for final evaluation. To ensure fair comparison, we employ standard grid search to select the best hyperparameters and model configurations for each method based on validation set performance. Due to the high computational costs involved, we conduct only one run for the segmentation experiments. However, our preliminary results indicate that segmentation results are insensitive to the choice of random seeds.

- Classification: We adhere to the approach followed by most existing methods, which involves adding a linear classifier on top of the pre-trained image encoder. Both the image encoder and the classifier are fine-tuned on each dataset. We use the binary cross entropy loss for multi-label classification and the cross entropy loss for multi-class classification. We set the maximum training epoch to 200. During grid search, we explore a large search space of hyper-parameters by selecting the learning rate form $\{3 \times 10^{-2}, 1 \times 10^{-2}, 3 \times 10^{-3}, 1 \times 10^{-3}, 5 \times 10^{-4}, 1 \times 10^{-4}, 5 \times 10^{-5}, 1 \times 10^{-5}\}$, the batch size from $\{32, 64, 128\}$, the optimizer from $\{SGD, Adam\}$, and whether custom refinements including Layer Normalization (LN) and Discriminative Learning Rates (DLR) discussed in Section 3.2 are applied or not.

- Segmentation: We adapt the UperNet architecture [40] based on the implementation provided by the open-source mmsegmentation package [8]. We fine-tune UperNet with a frozen backbone from the pre-trained MedVLP image encoder. To incorporate the segmentation head, we only make minimal modifications to ensure that the dimensions of the pre-trained image encoder and the UperNet network match for each method. Following the recommended settings of mmsegmentation, we utilize the cross-entropy loss for training and SGD as the optimizer with a momentum of 0.9 and a polynomial decay schedule. We set the maximum number of training iterations to 20,000 and the batch size to 32. The best learning rate is selected from $\{1 \times 10^{-2}, 1 \times 10^{-3}, 1 \times 10^{-4}\}$ for each dataset.

- Report Generation: We adapt R2Gen [4] as the task-specific head for report generation, with the image encoder frozen from a specified MedVLP model. Following the settings of R2Gen, we train the model using cross-entropy loss and the Adam optimizer. The maximum training epoch is set to 100, and the batch size is set to 16. We select the best learning rate from $\{1 \times 10^{-2}, 1 \times 10^{-3}, 1 \times 10^{-4}\}$.

- Image-Text Retrieval: We follow the same setting of CLIP-based VLP method to obtain the image and text embeddings from their respective pre-trained models. Subsequently, we compute the cosine similarity between a query image and all candidate reports to identify the target reports.

Table 10: Selected hyper-parameters per method on the VinDr dataset.

| Method | Learning Rate | Batch Size | Optimizer | LN | DLR |
|---|---|---|---|---|---|
| ConVIRT | $5 \times 10^{-5}$ | 32 | Adam | Yes | Yes |
| GLoRIA | $1 \times 10^{-4}$ | 64 | Adam | Yes | Yes |
| MedCLIP-R50 | $1 \times 10^{-4}$ | 128 | Adam | No | No |
| MedCLIP-ViT | $1 \times 10^{-4}$ | 128 | Adam | No | No |
| MedKLIP | $1 \times 10^{-4}$ | 64 | Adam | No | Yes |
| M-FLAG | $1 \times 10^{-4}$ | 64 | Adam | Yes | No |
| MGCA-R50 | $5 \times 10^{-5}$ | 64 | Adam | Yes | No |
| MGCA-ViT | $3 \times 10^{-2}$ | 64 | SGD | Yes | Yes |
| MRM | $1 \times 10^{-2}$ | 64 | SGD | Yes | Yes |
| REFERS | $3 \times 10^{-2}$ | 128 | SGD | Yes | No |

Table 11: Selected hyper-parameters per method on the COVIDx dataset.

| Method | Learning Rate | Batch Size | Optimizer | LN | DLR |
|---|---|---|---|---|---|
| ConVIRT | $5 \times 10^{-4}$ | 64 | Adam | Yes | Yes |
| GLoRIA | $5 \times 10^{-4}$ | 32 | Adam | Yes | Yes |
| MedCLIP-R50 | $5 \times 10^{-4}$ | 64 | Adam | No | No |
| MedCLIP-ViT | $1 \times 10^{-4}$ | 64 | Adam | No | No |
| MedKLIP | $1 \times 10^{-4}$ | 64 | Adam | No | Yes |
| M-FLAG | $5 \times 10^{-4}$ | 128 | Adam | Yes | No |
| MGCA-R50 | $5 \times 10^{-4}$ | 128 | Adam | Yes | No |
| MGCA-ViT | $5 \times 10^{-4}$ | 32 | Adam | Yes | Yes |
| MRM | $5 \times 10^{-4}$ | 64 | Adam | Yes | Yes |
| REFERS | $5 \times 10^{-4}$ | 64 | Adam | Yes | No |

Table 12: Selected hyper-parameters per method on the SIIM dataset.

| Method | Learning Rate | Batch Size | Optimizer | LN | DLR |
|---|---|---|---|---|---|
| ConVIRT | $1 \times 10^{-4}$ | 128 | Adam | Yes | Yes |
| GLoRIA | $1 \times 10^{-5}$ | 128 | Adam | Yes | Yes |
| MedCLIP-R50 | $1 \times 10^{-5}$ | 128 | Adam | No | No |
| MedCLIP-ViT | $1 \times 10^{-5}$ | 32 | Adam | No | No |
| MedKLIP | $1 \times 10^{-4}$ | 64 | Adam | No | Yes |
| M-FLAG | $1 \times 10^{-4}$ | 64 | Adam | Yes | No |
| MGCA-R50 | $1 \times 10^{-5}$ | 128 | Adam | Yes | No |
| MGCA-ViT | $1 \times 10^{-2}$ | 128 | SGD | Yes | Yes |
| MRM | $1 \times 10^{-2}$ | 64 | SGD | Yes | Yes |
| REFERS | $3 \times 10^{-2}$ | 64 | SGD | Yes | No |

Table 13: Selected hyper-parameters per method on the RSNA dataset.

| Method | Learning Rate | Batch Size | Optimizer | LN | DLR |
|---|---|---|---|---|---|
| ConVIRT | $5 \times 10^{-5}$ | 64 | Adam | Yes | Yes |
| GLoRIA | $1 \times 10^{-4}$ | 32 | Adam | Yes | Yes |
| MedCLIP-R50 | $1 \times 10^{-5}$ | 32 | Adam | No | No |
| MedCLIP-ViT | $1 \times 10^{-5}$ | 32 | Adam | No | No |
| MedKLIP | $1 \times 10^{-4}$ | 128 | Adam | No | Yes |
| M-FLAG | $1 \times 10^{-4}$ | 64 | Adam | Yes | No |
| MGCA-R50 | $1 \times 10^{-5}$ | 32 | Adam | Yes | No |
| MGCA-ViT | $1 \times 10^{-2}$ | 32 | SGD | Yes | Yes |
| MRM | $1 \times 10^{-2}$ | 32 | SGD | Yes | Yes |
| REFERS | $1 \times 10^{-2}$ | 32 | SGD | Yes | No |

### A.3 Selected Hyper-Parameters

In this section, we provide the selected hyper-parameters per method and dataset.

- Classification: Tables 9, 10, 11, 12, 13 show the selected hyper-parameters per method and dataset.

- Segmentation: When the pre-trained image encoder is frozen, we find the hyper-parameters are consistent in terms of the MedVLP methods. As a result, we select $lr = 1 \times 10^{-4}$ for Object CXR, $lr = 1 \times 10^{-4}$ for RSNA, $lr = 1 \times 10^{-3}$ for SIIM, and $lr = 1 \times 10^{-3}$ TBX11K.

- Report Generation: Similar to the segmentation experiments, the hyper-parameters are consistent in terms of the MedVLP methods. We find $lr = 1 \times 10^{-3}$ is the best learning rate for the IU X-ray dataset.

