# OpenReview forum: "BenchX: A Unified Benchmark Framework for Medical Vision-Language Pretraining on Chest X-Rays"
_NeurIPS.cc/2024/Datasets_and_Benchmarks_Track — NeurIPS 2024 Track Datasets and Benchmarks Poster_

### Official Review · Reviewer_CLKA · 2024-07-23
**The paper presents a significant and innovative benchmark framework that will promote progress in the MedVLP field.**

**Rating:** 8
**Confidence:** 4

**Review:**

1. The paper is well-organized and follows a logical structure, making it easy to follow the authors' arguments and methodologies.
2. The language is clear and precise, with technical terms appropriately explained.
3. The figures and tables are well-designed and effectively complement the text, aiding in the understanding of the presented data and results.
4. The inclusion of nine datasets and four diverse medical tasks is a novel approach that enhances the framework's comprehensiveness and utility. The discovery that earlier methods can outperform recent ones with appropriate training strategies is an intriguing and original finding that challenges current assumptions in the field. While the framework itself is original, the paper could benefit from a more detailed discussion of how BenchX differs from or improves upon existing benchmark efforts in related fields.

**Strengths:**

1. The paper introduces a comprehensive and standardized benchmark framework, which is urgently needed in the MedVLP field. BenchX allows for systematic comparison of different MedVLP methods' performance, which is crucial for advancing this domain.
2. BenchX includes nine datasets and four medical tasks, covering classification, segmentation, report generation, and image-text retrieval. This broad coverage helps evaluate the generalization capabilities of MedVLP methods across different tasks.
3. The proposed unified fine-tuning protocol addresses compatibility issues in task adaptation for different MedVLP methods, making the comparisons fairer and more transparent.
4. Through BenchX, the authors find that appropriate training strategies can significantly enhance the performance of some MedVLP methods, suggesting that reported results in current research may not reflect these methods' optimal performance. This finding prompts a reevaluation of existing methods' effectiveness and conclusions.

**Additional Feedback:**

There is no further feedback.

**Clarity:**

The paper is well-organized and follows a logical structure, making it easy to follow the authors' arguments and methodologies.

**Correctness:**

The evaluation methods and experiment design are appropriate and performed correctly.

**Documentation:**

Yes, the supplementary materials support the reproducibility.

**Ethics:**

There are no ethical concerns.

**Limitations:**

1. While the framework itself is original, the paper could benefit from a more detailed discussion of how BenchX differs from or improves upon existing benchmark efforts in related fields.
2. While the mDice score is a standard metric, providing a more detailed analysis of the segmentation results, such as visual examples or confusion matrices, could further strengthen the evaluation.
3. The paper mentions the potential for combining MedVLP models with large language models (LLMs) for report generation. Future work in this direction would be of interest to the community.

**Opportunities For Improvement:**

1. While the framework itself is original, the paper could benefit from a more detailed discussion of how BenchX differs from or improves upon existing benchmark efforts in related fields.
2. The paper could further emphasize the potential impact of BenchX on clinical practice, such as how improved MedVLP methods could translate into better diagnostic tools or more accurate medical reports.

**Relation To Prior Work:**

the authors find that appropriate training strategies can significantly enhance the performance of some MedVLP methods. This finding prompts a reevaluation of existing methods' effectiveness and conclusions.

**Summary And Contributions:**

This paper proposes a unified benchmark framework called BenchX for head-to-head comparison and systematic analysis of medical vision-language pretraining (MedVLP) methods. BenchX aims to address discrepancies in datasets, preprocessing, and fine-tuning implementations among existing MedVLP methods, thus enabling fair and rigorous evaluation. The paper thoroughly describes the three main components of BenchX: the comprehensive dataset, the benchmark suite, and the unified fine-tuning protocol. Through BenchX, the authors establish baselines for nine state-of-the-art MedVLP methods and discover that some earlier methods can surpass recent ones with appropriate training strategies. This finding prompts a reevaluation of previous research outcomes and conclusions.

---

> ### Author Rebuttal · Authors · 2024-08-16
>
> **To Reviewer CLKA:**
>
> **Q3.1**
> The paper could benefit from a more detailed discussion of how BenchX differs from or improves upon existing benchmark efforts in related fields
>
> **A3.1**
> We discuss the differences between our BenchX framework and existing benchmarks in response **A0.2** to all the reviewers.
>
> **Q3.2** The paper could further emphasize the potential impact of BenchX on clinical practice, such as how improved MedVLP methods could translate into better diagnostic tools or more accurate medical reports.
>
> **A3.2** Thank you for the suggestion. BenchX has the potential to significantly advance AI-driven medical tools, leading to more reliable and accurate clinical outcomes:
>
> 1. **Improved Diagnostic Tools:** By identifying the most effective MedVLP models, BenchX can enhance diagnostic tools, helping healthcare professionals reduce misdiagnoses and provide more precise care.
>
> 2. **Accelerated AI Development:** BenchX serves as a robust framework with standardized datasets and protocols, streamlining the development of new medical AI models. This contributes to innovation and helps in the development of advanced diagnostic tools.
>
> 3. **Support for Clinical Research:** BenchX offers a consistent benchmark for evaluating new methods, supporting clinical research and facilitating the quicker application of research findings in practice, ultimately leading to better patient care.
>
> We will add the above discussion in the final version of our paper.

---

### Official Review · Reviewer_magQ · 2024-07-24
**Comprehensive benchmark work but lacks significant novelty in this field. But still could be accepted for the high quality benchmark.**

**Rating:** 6
**Confidence:** 4
**Correctness:** N/A
**Clarity:** N/A

**Review:**

Extensive experiments but lacks novelty in this field, Benchmark framework on Chest X-ray, which covers a comprehensive range.

**Strengths:**

1. As shown in Tab. 1,2, there are extensive baseline models tested and benchmarked.
2. This paper covers a wide range of downstream task including diagnosis, report generation, etc.

**Additional Feedback:**

N/A

**Documentation:**

N/A

**Opportunities For Improvement:**

1. Include the benchmark of SOTA multi-modal LLMs.
2. Use a seperate subsection to describe the differences with similar work.

**Relation To Prior Work:**

N/A

**Summary And Contributions:**

The paper discusses the potential of Medical Vision-Language Pretraining (MedVLP) in generating robust visual representations from medical images and reports, which are crucial for enhancing the adaptability of task-specific models with minimal examples. However, due to inconsistencies in datasets, preprocessing, and finetuning across existing MedVLP methods, a standardized benchmark, BenchX, is proposed to facilitate comprehensive comparisons and analyses using public chest X-ray datasets.

Contributions:

1. Introduction of BenchX, a unified benchmark framework designed to enable systematic and direct comparisons between various MedVLP methods, addressing the challenge of non-standardized evaluations in the field.
2. BenchX comprises comprehensive datasets, standardized benchmark suites for processing and evaluation, and unified finetuning protocols, supporting consistent adaptation across classification, segmentation, and report generation tasks.
3. The application of BenchX revealed that properly optimized earlier MedVLP methods could perform comparably or better than more recent approaches, suggesting a need to reassess previous advancements and conclusions in the MedVLP research domain.

---

> ### Author Rebuttal · Authors · 2024-08-16
>
> **To Reviewer magQ:**
>
> **Q2.1** Include the benchmark of SOTA multi-modal LLMs.
>
> **A2.1** We conduct preliminary experiments to test multi-modal LLMs on our benchmark. Please refer to response **A0.1** to all the reviewers for more details.
>
> **Q2.2** Use a separate subsection to describe the differences with similar work.
>
> **A2.2** Thank you for the suggestion. We discuss the differences between our BenchX framework and existing benchmarks in response **A0.2** to all the reviewers, which will be added as a separate subsection in the final version of the paper.

---

### Official Review · Reviewer_LeEn · 2024-07-27
**BenchX: A Unified Benchmark Framework for Medical Vision-Language Pretraining on Chest X-Rays**

**Rating:** 6
**Confidence:** 4
**Correctness:** N/A
**Clarity:** N/A

**Review:**

**Advantages:**

- The proposed benchmark includes various tasks, such as Medical Image Classification, Medical Image Segmentation, Radiology Report Generation, and Medical Image-Text Retrieval.
- The paper conducts extensive experiments on the proposed benchmark, providing a comprehensive evaluation.


**Weaknesses:**

- The proposed benchmark focuses on the evaluation of the image encoder, which limits its ability to evaluate decoder-only multimodal LLMs.
- Due to the characteristics of medical data, appropriate human evaluation is needed to confirm the effectiveness of the benchmark.
- Although the benchmark standardizes data preprocessing, train-test splits, and parameter selection, all the datasets used in this paper are publicly available, without any further evaluation of data quality. The paper should conduct additional experiments to ensure the quality of the data.
- The paper could benefit from a table summarizing the average results across different tasks to provide an overall performance metric.

**Strengths:**

- The proposed benchmark includes various tasks, such as Medical Image Classification, Medical Image Segmentation, Radiology Report Generation, and Medical Image-Text Retrieval.
- The paper conducts extensive experiments on the proposed benchmark, providing a comprehensive evaluation.

**Additional Feedback:**

N/A

**Documentation:**

N/A

**Opportunities For Improvement:**

- The proposed benchmark focuses on the evaluation of the image encoder, which limits its ability to evaluate decoder-only multimodal LLMs.
- Due to the characteristics of medical data, appropriate human evaluation is needed to confirm the effectiveness of the benchmark.
- Although the benchmark standardizes data preprocessing, train-test splits, and parameter selection, all the datasets used in this paper are publicly available, without any further evaluation of data quality. The paper should conduct additional experiments to ensure the quality of the data.
- The paper could benefit from a table summarizing the average results across different tasks to provide an overall performance metric.

**Relation To Prior Work:**

N/A

**Summary And Contributions:**

This paper proposes a unified benchmark framework that enables head-to-head comparison and systematic analysis between MedVLP methods using 11 public chest X-ray datasets.

---

> ### Author Rebuttal · Authors · 2024-08-16
>
> **To Reviewer LeEn:**
>
> **Q1.1** The proposed benchmark focuses on the evaluation of the image encoder, which limits its ability to evaluate decoder-only multimodal LLMs.
>
> **A1.1** We acknowledge that one limitation of our work is the lack of benchmarking for medical multimodal LLMs, as discussed in Section 4.7. Given that MedVLP is crucial for training medical multimodal LLMs, our BenchX framework could be valuable for evaluating which MedVLP methods are most effective for building medical multimodal LLMs. This could be further explored by extending the downstream tasks to LLM-based report generation and VQA, which we plan to address in future work. For comprehensiveness, we conduct preliminary experiments to test multimodal LLMs. Please refer to the response **A0.1** to all the reviewers for more details.
>
> **Q1.2** On human evaluation and data quality.
>
> **A1.2** We understand your concern about data quality. Our primary reason for testing exclusively on public datasets is to ensure that benchmarking is both reproducible and transparent. The public datasets used in our BenchX framework were originally collected and annotated by clinical experts from reputable hospitals. These datasets have been widely adopted to evaluate the performance of medical AI models on chest X-rays.
>
> We believe that the datasets we compared are of high quality. To confirm this, we had experienced radiologists review the test sets of the compared datasets, and they verified that the quality of the test data is sufficient for benchmarking MedVLP methods.
>
> **Q1.3** The paper could benefit from a table summarizing the average results across different tasks to provide an overall performance metric.
>
> **A1.3** Thanks for the suggestions. Table 2 shows the overall performance of each MedVLP method across various tasks including multi-label classification (M-CLS), binary classification (B-CLS), segmentation (SEG), and radiology report generation (RRG). The experimental results indicate that MRM and MGCA-ViT generally outperform the other methods. We will add the above mentioned results in the final version of our paper.
>
> **Table 2. Overall performance of each MedVLP method across different tasks.**
>
> | Method         | M-CLS (AUC) | B-CLS (F1) | SEG (mDice) | RRG (BLEU4) | Avg    | Rank |
> |----------------|:-----------------------:|:-----------------:|:-------------:|:-------------:|:--------:|------:|
> | ConVIRT        | 85.37                 | 65.56           | 78.89       | 0.148       | 57.492 | 4    |
> | GLoRIA         | 84.68                 | 64.06           | 77.05       | 0.170       | 56.49  | 9    |
> | MedCLIP-R50    | 83.02                 | 67.17           | 79.80       | 0.163       | 57.038 | 7    |
> | MedCLIP-ViT    | 84.00                 | 68.33           | 78.76       | 0.151       | 57.31  | 5    |
> | MedKLIP        | 82.77                 | 65.56           | 79.42       | 0.167       | 56.98  | 8    |
> | M-FLAG         | 77.73                 | 62.96           | 72.77       | 0.147       | 53.902 | 10   |
> | MGCA-R50       | 83.47                 | 64.69           | 79.85       | 0.159       | 57.042 | 6    |
> | MGCA-ViT       | 86.10                 | 67.03           | 80.32       | 0.170       | 58.405 | 2    |
> | MRM            | 86.18                 | 67.72           | 80.66       | 0.165       | 58.681 | 1    |
> | REFERS         | 84.65                 | 66.06           | 79.93       | 0.161       | 57.7   | 3    |

---

### Author Rebuttal · Authors · 2024-08-16

**To all the reviewers:**

We thank the insightful and valuable comments from all the reviewers. In what follows, we will first provide overall responses to the common comments and then address the specific concerns from each reviewer individually:

**Q0.1** On the evaluation of multimodal LLMs.

**A0.1** We conducted preliminary experiments to assess the performance of multimodal LLMs on our benchmark for comprehensiveness. Due to limited time, we only test LLaVA [1] and LLaVA-Med [2] with LoRA instruct-tuning on the NIH and VinDr datasets.

As shown in Table 1, while LLaVA and LLaVA-Med with naive fine-tuning exhibit some diagnostic capability, they still fall short compared to task-specific MedVLP models. This highlights significant room for improvement in the use of multimodal LLMs for diagnostic purposes.

**Table 1. Classification results of multimodal LLMs on the NIH and VinDr datasets.**

| Dataset | Method         | Acc  | F1   | AUC  |
|---------|----------------|------|------|------|
| NIH     | LLaVA1.5-13B   | 0.521| 0.028| 0.506|
|         | LLaVA-Med-7B   | 0.416| 0.054| 0.508|
| VinDr   | LLaVA1.5-13B   | 0.651| 0.159| 0.581|
|         | LLaVA-Med-7B   | 0.605| 0.148| 0.569|

**Q0.2** On the differences between the proposed BenchX framework and existing works.

**A0.2** BenchX distinguishes itself from existing benchmarks like TorchXRayVision and ViLMedic by addressing key gaps in the evaluation and comparison of MedVLP methods:

1. **Comprehensive Multimodal Benchmarking:** Unlike TorchXRayVision, which is limited to vision tasks, BenchX offers a unified framework that benchmarks MedVLP methods across both vision and language tasks, facilitating the evaluation of models that integrate multimodal data.

2. **Standardized Task Adaptation Protocol:** BenchX introduces a consistent and standardized task adaptation pipeline, addressing a gap in ViLMedic. This ensures fair comparisons across different MedVLP methods by minimizing variability due to differing implementation details.

3. **Extensive Evaluation Scope:** BenchX surpasses ViLMedic by evaluating a broad range of MedVLP methods across multiple downstream tasks. This allows for a more thorough comparison, providing deeper insights into the strengths and weaknesses of each method.

4. **Diverse Dataset and Benchmark Suite:** BenchX includes a comprehensive dataset and diverse benchmark suite, addressing limitations in existing frameworks. This enables a more robust and reliable evaluation of MedVLP methods, establishing new baselines and promoting advancements in the field.


[1] Liu, H., Li, C., Wu, Q., & Lee, Y. J. (2024). Visual instruction tuning. *Advances in Neural Information Processing Systems, 36.*

[2] Li, C., Wong, C., Zhang, S., Usuyama, N., Liu, H., Yang, J., Naumann, T., Poon, H., & Gao, J. (2024). Llava-med: Training a large language-and-vision assistant for biomedicine in one day. *Advances in Neural Information Processing Systems, 36.*

---

### Decision · Program_Chairs · 2024-09-26

**Decision:**

Accept (Poster)

**Comment:**

This submission received three ratings (6, 6, and 8), averaging 6.67, which is a positive score. The reviewers provided some suggestions about analysis, metric and future directions, which has been considerably demonstrated during the rebuttal. After rebuttal, it seems that no reviewers have further questions and after I carefully double check the rebuttal contents, I found the evidence can be well supported the improvement. Overall, I suggested the acceptance given the positive scores and hope the authors finally incorporate the suggestions by reviewers to improve the manuscript.